# Procurement process and shortages of essential medicines in public health facilities: A qualitative study from Nepal

Basant Adhikari[1,2,3☯], Kamal Ranabhat[4,5☯], Pratik Khanal[2,6☯], Manju Poudel[7], Sujan Babu Marahatta[8,9,10,11], Saval Khanal[12], Vibhu Paudyal[13,14], Sunil Shrestha[15☯]*

1 Ministry of Health, Bagmati Province, Hetauda, Nepal, 2 Nepal Public Health Association, Lalitpur, Nepal, 3 ASEAN Institute for Health Development, Mahidol University, Salaya, Thailand, 4 Ministry of Health and Population, Government of Nepal, Kathmandu, Nepal, 5 Central Department of Public Health, Institute of Medicine, Tribhuwan University, Kathmandu, Nepal, 6 Department of Global Public Health and Primary Care, Bergen Centre for Ethics and Priority Setting in Health, University of Bergen, Bergen, Norway, 7 Oxford University Clinical Research Unit, Patan Academy of Health Science, Lalitpur, Nepal, 8 Nepal Open University, Lalitpur, Nepal, 9 Manmoohan Memorial Institute of Health Science, Kathmandu, Nepal, 10 Department of Public Health Sciences, University of California Davis, Davis, California, United States of America, 11 Bournemouth University, Poole, United Kingdom, 12 Health Economics Consulting, Norwich Medical School, University of East Anglia, Norwich, United Kingdom, 13 School of Pharmacy, College of Medical and Dental Sciences, University of Birmingham, Birmingham, United Kingdom, 14 Florence Nightingale Faculty of Nursing, Midwifery and Palliative Care, King's College, London, United Kingdom, 15 School of Pharmacy, Monash University Malaysia, Selangor, Malaysia

☯ These authors contributed equally to this work.
* sunilcresta@gmail.com

**Data Availability Statement:** All the data underlying the findings are available within the manuscript itself.

## Abstract

Ensuring access to essential medicines remains a formidable challenge in Nepal. The specific reasons for the shortage of essential medicines within Nepal have not been extensively investigated. This study addresses challenges associated with access to essential medicines, procurement process difficulties, and functionality of inventory management systems at different levels of public health facilities. Fifty-nine semi-structured in-depth interviews were conducted with health managers and service providers at provincial and local levels in six randomly selected districts of Bagmati province, Nepal. Interviews were audiotaped and transcribed verbatim, and the results were analyzed using the inductive approach and were later mapped within the four domains of "Procurement of essential medicines". The major barriers for the effective management of essential medicines included delays in the procurement process, primarily locally, leading to frequent stock-out of essential drugs, particularly at the health post level. Additionally, challenges arise from storage problems, mainly due to insufficient storage space and the need to manage additional comorbidities related to COVID-19. Other identified challenges encompass the absence of training on logistics management information systems, a lack of information technology resources in primary health facilities, inadequate qualified human resources to operate the IT system, and insufficient power backup. Moreover, unrealistic demand estimation from the service points, inadequate transportation costs, and manual inventory management systems further contributed to the complex landscape of challenges. This study identified procurement delays as the primary cause of essential medicine shortages in Bagmati Province, Nepal. We recommend

**Funding:** The Health Logistics Management Center, Ministry of Social Development, Bagmati Province funded the study. The funding body did not play any role in the study design, collection, analysis, and interpretation of data, nor in writing this menu script. However, no specific funding was received for the publication of this work.

**Competing interests:** No

implementing comprehensive procurement guidelines, collaborative training, and dedicated budgets to address this issue. Improving the procurement and inventory management process in low-resource settings requires a well-trained workforce, suitable storage spaces, and enhanced coordinated administrative tiers within health facilities at different levels to ensure the year-round availability of essential medicines in these settings.

## Introduction

The effective management of pharmaceutical supply chains is indispensable for delivering quality healthcare services. Challenges such as shortages of medicines, stock-outs of essential medicines, procurement processes, and inventory management systems (IMS) persist in delivering quality healthcare services globally, mainly in LMICs like Nepal [1, 2]. While acknowledging the global prevalence of studies on stock-outs [3, 4], barriers [5–7], and supply chain issues [8, 9] related to essential medicines, it is necessary to highlight the unique circumstances faced by the health system of Nepal. Despite the abundance of worldwide research in this area, the focus on Nepal remains limited, creating a gap in understanding the specific challenges within its context.

The complexity of the healthcare supply chain involves the coordination of diverse stakeholders, including health managers, service providers, pharmaceutical manufacturers, distributors, information service providers, regulatory agencies, and service users [10]. Any disruptions in medicines supplies and IMS in health facilities have a broad and direct impact on the healthcare provision and patients' treatment process, especially in the aftermath of some natural disasters or human-made disasters. Over the last decades, researchers and practitioners have conducted numerous studies addressing essential medicines and IMS supplies, exploring their vulnerabilities to systemic problems in health facilities and emergencies [11–14].

In recent years, Nepal, classified as a lower-middle-income country by the World Bank, has undergone major political restructuring, transitioning to a federal system of governance. This decentralization has conferred greater responsibility upon provincial and local governments to provide quality health services, making effective logistics management essential [15]. However, the health system in Nepal, particularly at sub-national levels, faces persistent challenges in maintaining the year-round availability of drugs and supplies, exacerbated during public health emergencies. While local health facilities offer 70 essential drugs and 15 medical and surgical devices free of charge, ongoing concerns about their availability persist despite government investments in procurement and distribution. The public and the media frequently bring these concerns to attention during program reviews [16]. Disruption in the regular supply of pharmaceutical products, including essential drugs in the public sector, is multifactorial, impacting the government's goal of ensuring universal access to essential medicines [17–19]. A critical gap exists in the systematic analysis of essential drug distribution systems and their impact on stock-out rates at the point of service delivery in Nepal [20]. Limited studies have investigated the factors causing stock-outs of essential products at the point of service delivery [1, 21–23]. This study aims to bridge the gap by exploring the reasons for shortages and stock-outs, challenges in the procurement process, and IMS in health facilities/medical stores in Nepal. By providing a subtle understanding of the issues specific to Nepal, this research contributes valuable insights for enhancing pharmaceutical supply chain management in LMICs.

## Methods

### Study design and study sites

The study adopted a qualitative research method, which included in-depth interviews with service providers and health managers working in health facilities and local/provincial governments of Bagmati province. Bagmati is one of the seven provinces of Nepal and has 13 districts. During the survey, the Ministry of Social Development (MoSD) was the focal ministry for health, while the Health Logistics Management Center (HLMC) under the MoSD was responsible for overseeing the management of medicines, health equipment and supplies. Ministry of Health, a newly formed ministry, oversees the province government's health functions. **Fig 1** shows the healthcare supply chain cycle in Nepal.

### Sampling procedure and sample size

We randomly selected six out of 13 districts in the Bagmati province, namely Chitwan, Nuwakot, Lalitpur, Sindhupalchowk, Makwanpur and Sindhuli. Kathmandu and Bhaktapur districts were excluded from the sampling frame as they lacked rural municipalities. Exclusion criteria were applied to districts lacking a rural municipality, emphasizing the intentional inclusion of districts with both municipality and rural municipality setups. This strategic approach assures a broader representation of diverse settings in our study sites, supporting our objective of comprehensively understanding the investigated factors. A provincial hospital and health office were selected from each district. The health office is a unit of the province functioning at the

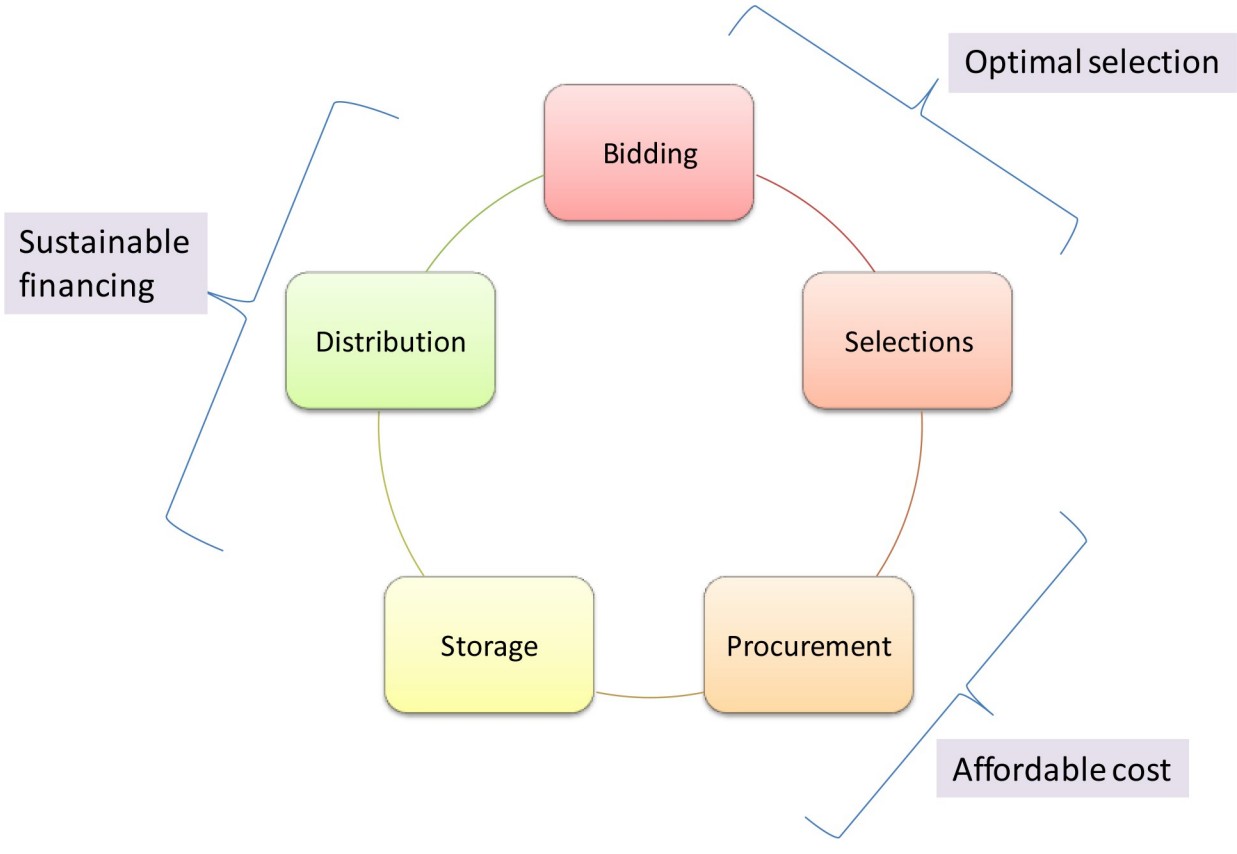

**Fig 1. Healthcare supply chain cycle.**

**Table 1. List of the district and local governments selected for the study.**

| District/HLMC | Municipality/Metro | Rural Municipality | Number of in-depth interviews |
|---|---|---|---|
| Chitwan | Bharatpur Metropolitan City | Ichchhakamana Rural Municipality | 10 |
| Makwanpur | Hetauda Sub-Metropolitan City | Bagmati Rural Municipality | 8 |
| Lalitpur | Lalitpur Metropolitan City | Mahankal Rural Municipality | 10 |
| Sindhuli | Dudhauli Municipality | Sunkoshi Rural Municipality | 10 |
| Nuwakot | Bidur Municipality | Kispang Rural Municipality | 10 |
| Sindhupalchowk | Chautara Sangachowkgadi | Tripurasundari Rural Municipality | 10 |
| HLMC | - | - | 1 |
| Total | | | 59 |

district level, replacing the district (public) health offices during pre-federalism in Nepal. The researchers interviewed the store in charge of the province hospital and the chief or store in charge of health offices.

Two local governments (one metro/municipality and one rural municipality) were then randomly selected from each of the six districts (n = 12 local governments). These local governments were either metropolitan, sub-metropolitan, municipalities or rural municipalities depending on the country's administrative regulations, mostly indicating their development status. Local governments have health departments or sections that govern primary hospitals, primary health centers (PHCs), and health posts. Each local government's health coordinator/ health section chiefs participated in the interviews.

Similarly, one PHC and two health posts were selected from each of the 12 local governments and the people in charge of each were interviewed. However, interviews with Hetauda sub-metropolitan city health coordinators and Bagmati rural municipality could not be conducted as the officials were unavailable. The reason was that one of the officials opted for an in-person interview, which was not possible due to the movement required during COVID-19, while the other could not be approached due to poor mobile network communication despite multiple contacts. We also conducted an in-depth interview with the logistics personnel of HLMC. A total of 59 in-depth interviews out of 61 planned were conducted. Table 1 shows the list of the district and local governments selected for the study.

## Data collection methods

In-depth interviews were carried out with personnel involved in the logistics management of the organization/health facility. For the interview purpose, an interview guide was developed comprising socio-demographic information of the study participants, questions related to the availability of essential medicines, product selection, quantification and procurement of medicines, demand-supply chain, storage and distribution, and IMS. Additionally, open-ended questions were added to the interview guide to understand participants' perspectives on the scope for improvement.

Data collection was undertaken by six research assistants (public health undergraduates) and one supervisor in June-July 2021. Study participants were interviewed through Zoom virtual/Google Meet platform and telephone as the study was conducted during the COVID-19 pandemic. Physical interaction with the study participants was only done in the Sindhupalchowk district. Training was provided to the interviewers by the research team and programmatic managers of the HLMC of Bagmati province. The training duration was three days, covering the basics of logistic management at the province level, orientation on data collection tools and consent-taking processes, communication skills, coordination approach, and strategies for recording and

transcribing the audio transcripts. After completing the interview process, study members cross-checked for compliance with the interview guide (Available in S1 Text).

## Data management and analysis

The audio recordings were first transcribed and translated into the English language. Thematic analysis was undertaken to interpret the domain, themes and patterns using the inductive analysis approach. First, codes were identified by familiarizing with the transcripts and highlighting the significant statements made by the study participants. These codes were grouped into themes and later categorized under domains. The quality of the transcripts was assessed by verifying the audio records and the transcripts by authors (KR and PK). Similarly, the independent research member (MP and SS) verified the codes and themes generated.

## Ethical approval

Administrative approval for data collection was obtained from the HLMC, Bagmati province. Ethical approval was obtained from the Ethical Review Board of Nepal Health Research Council (Reference number: 249/2021 P). Informed consent was obtained from the participants, both written and verbal, and recorded before conducting the interviews. Due to the COVID-19 pandemic, the predominant method for obtaining informed consent was either through verbal consent or through email, utilizing digital signatures for authentication. Additionally, a limited number of consents were gathered in person, ensuring strict adherence to COVID-19 protocols at that time. The anonymity and confidentiality of the respondents were maintained throughout the study. Only the research team had access to the transcripts of the interviews.

# Results

## Characteristics of study population

Fifty-nine individuals participated in the study, including health workers, health coordinators, managers, and storekeepers. The designation of health workers participating in the study is shown in Table 2.

## Overarching domain

Data analysis showed the four overarching domains: (i) purchasing of essential medicine, (ii) inventory management, (iii) demand and supply of essential medicines, and (iv) shortage of essential medicines (see **Fig 2**).

**Table 2. Characteristics of study participants.**

| Designation | Number (n = 59) | Percentage |
|---|---|---|
| Health Post In-charge (Public health inspector, Health assistant, Senior/Auxiliary Health Worker) | 34 | 57.6 |
| Health coordinator/Section chief (municipality, rural municipality, metropolitan) | 8 | 13.6 |
| Store in-charge/Admin chief, Hospital | 6 | 10.2 |
| Store in-charge, Health office | 4 | 6.8 |
| Chief, Health Office | 2 | 3.4 |
| PHC In-charge | 2 | 3.4 |
| Other municipal staff (computer operator, store personnel) | 2 | 3.4 |
| Logistics officer, HLMC | 1 | 1.6 |
| **Total** | **59** | **100** |

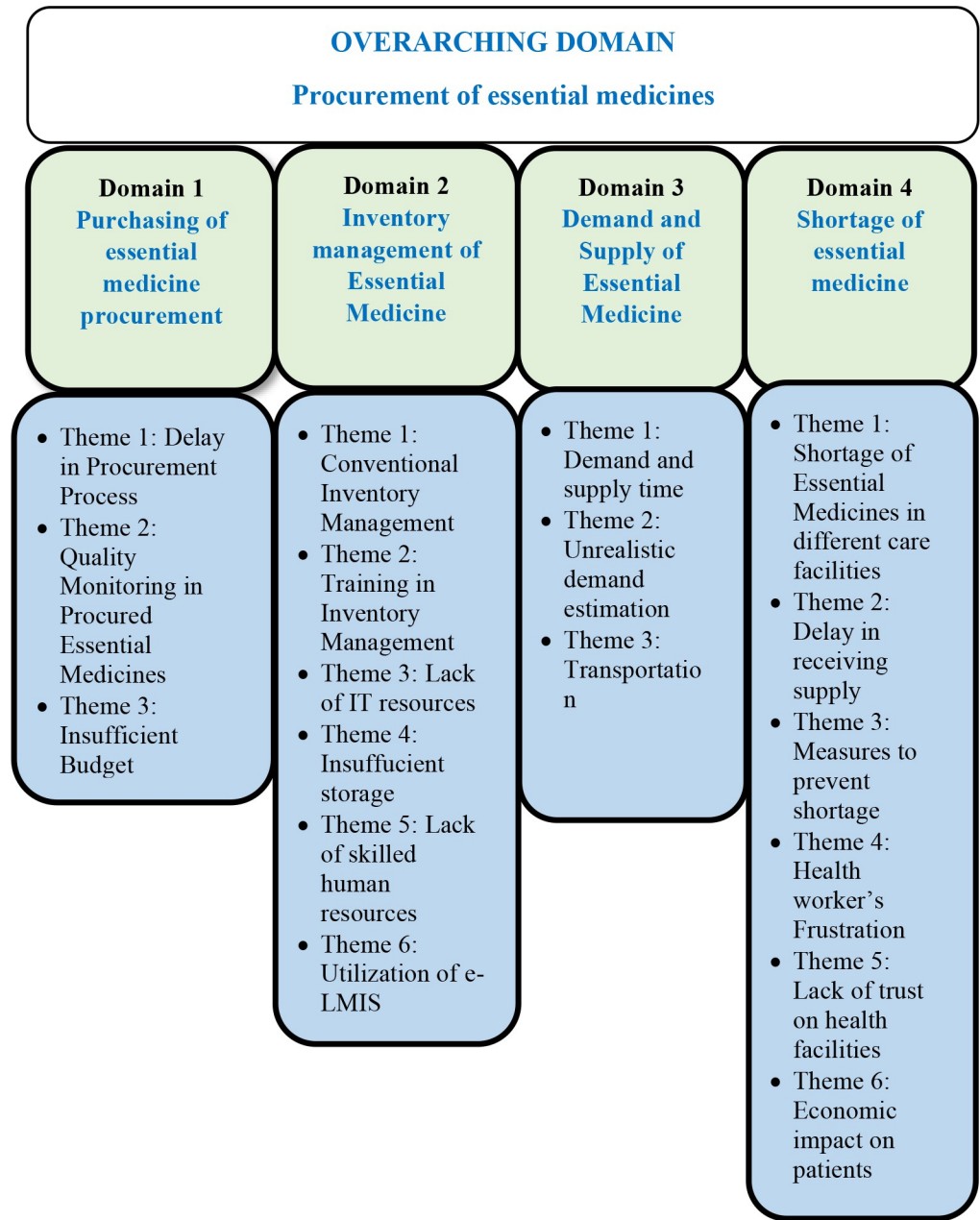

**Fig 2. Overarching domains in the procurement of essential medicines.**

## Domain 1: Purchasing of essential medicine

**Theme 1: Delay in the procurement process.** Participants universally perceived the procurement process as time-consuming, resulting in delays in the supply chain. The administrative process for procurement starts in the first quarter, and essential medicines reach service delivery points only in the third quarter.

Various stakeholders, including HLMC, province hospitals, and local governments, are involved in procurement. However, health offices and primary health centers/health posts are not engaged in the process. A tender is opened to procure essential medicines amounting to

while a quotation is called for procurement of essential medicines amounting. Quotes from Logistic Officer and Store In-charge highlight the challenges in the procurement process.

> "*Procurement process is time-consuming. So, we plan to initiate procurement in the first quarterr and distribute it in the second.*" (Logistic Officer, HLMC, Bagmati Province)

> "*It's a difficult process, we have to arrange different lists, and the process is long, we cannot reach all the suppliers for the rate and quotation, so we have to ask supplier in our roster for the drugs via phone. Medicine might get out of stock at any time.*" (Store In-charge, Province Hospital, Hetauda)

> *Due to the long process of procurement of medicines and delay in receiving the supply, we cannot provide essential medicines regularly for up to 2–3 months per year.* (Administrative head, Hospital, Lalitpur)

Most participants from health posts said that the procurement process from the local level was lengthy, leading to consistent delays. Consequently, health posts faced shortages of essential medicines.

> "*Tender process from the municipality starts late. We just reported about scarcity of medicines, and the municipality replied that the tender is being processed. We should follow the decision of the municipality.*" (Senior AHW, Health Post, Makwanpur)

**Theme 2: Quality monitoring in procured essential medicines.** Concerns were raised regarding the inadequate monitoring measures during the procurement process, leading to issues such as the supply of near-to-expiry or low-quality drugs.

Insights from participants' quotes illuminate the difficulties associated with quality monitoring.

> "*We usually are experiencing compromised standards of essential medicines in the tender process. Near expiry stocks of medicines, damaged product and even it seems, low-quality products are supplied.*" (Health Assistant, Rural Municipality, Lalitpur)

The lack of clarity in roles between province and local levels, coupled with insufficient budget allocations, exacerbates the risk of procuring substandard medicines.

> "*The tender document at the local government level does not specify shelf-life requirement. Thus, there could be essential medicines with a shelf life even for 12 to 14 months.*" (Logistic Officer, HLMC, Bagmati Province)

A health coordinator at the local government opined that focusing only on price during the tender process raises the issue of quality assurance in the supply of essential medicines at the local level.

> "*While only analyzing the price of essential medicines during the procurement process, we need to compromise quality. Quality checking provision should be made in each level.*" (Health Coordinator, Municipality, Sindhupalchowk)

According to a province official, E-LMIS reporting and First Expiry First Out (FEFO) rule can also help maintain the quality of essential medicines. However, inadequate storage space and supply of COVID-19 commodities have made it difficult to track the stock of essential medicines.

**Theme 3: Insufficient budget.** Despite the dedicated budget at the province, hospitals, and local governments, several participants expressed that the allocated funds were insufficient to ensure a year-round supply of essential medicines.

Quotes from participants highlight the budgetary constraints.

*"We don't have enough budget; the hospital management committee also does not have the budget. Due to lack of medicines, we cannot do some programs such as medical camps."* (Administrative and Logistics manager, Hospital, Sindhupalchowk)

An official from the health office suggested that the role of the health office at the district level should be strengthened in procuring essential medicines. Health offices currently only play the role of mediators in supply chain management.

*"Health office should also have authority in procurement. Health offices should be strengthened in their role in ensuring that there is no shortage of essential medicines and other supplies in health facilities."* (Health Office, Lalitpur)

A few participants at the health post level emphasized the need for budget allocation to address the immediate shortage of medicines.

'*We had a shortage of condoms and Depo (injectables) for the last three months. The shortage would have been managed if there was budget allocation at the health post level.*' (Health post, Sindhupalchowk)

Almost all health posts managed inventory through the conventional register method, while hospitals, primary health centers, and municipal offices used both computer-based and registry methods. Limited resources, such as computers and the internet, hindered effective implementation at the health post level. Participants from health posts said there was not enough space to record in the register, and it was hard to maintain the record in good condition. At most local levels, three-monthly reports were recorded in e-LMIS. Some hospitals and local levels recorded spendable items (reduced after use, such as medicines, liquid hand wash, etc.) through e-LMIS and non-spendable (such as refrigerators) through the register.

*"We record in register 'dhaddaa'. We don't have computer, and we are not trained on the online inventory management system."* (Health Post In-charge, Chitwan)

Some rural municipalities entered data in both registers and computers. Data was recorded in the register for cross-checking purposes and because the register was convenient for them.

*"Daily, the entry is done in the computer, and for cross-checking, we keep the record in record book 'dhadda'."* (Health Coordinator, Rural Municipality, Sindhuli)

Computer systems were used for inventory management at most local levels, hospitals, and health offices.

*"We use both systems. The computer system is used for spendable items and medicines on (Ma Lay Pa 52), and the registration system is used for non-spendable items and materials on (Ma Lay Pa 47)."* (Cold Chain Officer, Health Office, Lalitpur)

**Theme 2: Training in inventory management.** Training in e-LMIS was identified as inadequate, with resource constraints and changes in authority and responsibility contributing to the gap.

'*We are not trained in e-LMIS. Training is required for us to maintain efficiency and uniformity.*' (Auxiliary Nurse Midwife, Health Post, Sindhuli)

'*We have not received training in e-LMIS. E-LMIS and procurement-related training is required in Metropolitan cities to make the local level independent in purchasing medicines.* (Public Health Officer, Bharatpur Metropolitan City, Chitwan)

One of the participants from the hospital level had never heard about e-LMIS training.

Province government officials cited resource constraints and changes in the authority and responsibility for inadequate training at the health facility level.

"*Last year, it was the responsibility of HLMC for basic logistic training at the local government level (forecasting, quantification), but this year, it is the responsibility of the National Health Training Center. Province lacks the resources to provide training at the local level. Newly recruited health cadres have not received training.*" (Logistic Officer, Bagmati Province)

**Theme 3: Lack of IT resources.** The absence of computers and reliable internet connectivity in most health posts hampered the adoption of efficient inventory management practices.

"*In register "Dhadda", it is time-consuming. In an online computer system, sometimes software errors occur like command not accepting pops up. So, we must keep a record in both "dhadda" and computer.*" (Health Coordinator, Rural Municipality, Sindhuli)

"*We don't use register; we use the online system. But we have an internet problem.*" (Health Assistant, Municipality, Sindhuli)

**Theme 4: Insufficient storage space.** Most participants said adequate storage space is a massive problem in managing essential medicines. Storage problems were addressed by renting or managing space in the corridor. Insufficient storage space emerged as a significant challenge, compromising adherence to the FEFO. In a public health emergency like the COVID-19 pandemic, the extra supply of logistics/medicines in response to health emergencies creates insufficient storage.

"*We have a huge problem with the store. We do not have enough space, so we are managing hall to store essential medicines.*" (Administrative Head, Hospital, Lalitpur)

"*We have a lack of storage space. As a result, medicines are not stored in the standard rule of storekeeping. Newly procured medicines come in front, and old medicines remain in the back of the storeroom. So FEFO rule is difficult to practice. As a result, expired drugs are piled up in the storeroom.*" (Health Post In Charge, Chitwan)

A participant from Sindhupalchowk Hospital said that the Gorkha Earthquake in 2015 damaged the hospital's building in Nepal.

"*We lost our hospital's building in the earthquake 2015. We are in a temporary settlement. There is no good building to rent nearby. Due to limited space in the temporary settlement, storage space of medicines is lacking.*" (Administrative and Logistics manager, Hospital, Sindhupalchowk)

**Theme 5: Lack of skilled human resources.**    The majority of the participants lacked training in logistic management, resulting in suboptimal store management. The frequent transfer of trained healthcare providers is another challenge for managing essential drugs and the supply chain system.

"*Due to lack of training on logistic management, we do not have skilled human resources for store management.*" (Health Post In-charge, Chitwan)

Participants expressed concerns about being overburdened with multiple responsibilities, including cold chain management and recording/ reporting.

"*A single storeroom is covered with dumped waste materials and is hard to manage alone. I work as a store manager, cold chain manager, and as a recording and reporting focal point of COVID-19 in the health office; there is a lack of workforce for store management in our health office.*" (Cold Chain Officer, Health office, Lalitpur)

**Theme 6: Utilisation of e-LMIS.**    While e-LMIS is designed to predict the stock of drugs available and identify medicines' expiration dates, limited storeroom space hindered the implementation of the FEFO rule.

"*Though we can find the expiry dates of medicines through e-LMIS, it is difficult to practice the FEFO because we have limited space in the store. With every new purchase, old products go to the corner, and in the end, we cannot go to the corner to use near-expiry products.*" (Logistic Officer, HLMC, Bagmati province)

## Domain 3: Demand and supply of essential medicines

**Theme 1: Demand and supply time.**    Health facilities typically demand essential medicines on a three-monthly basis, adjusting based on the consumption pattern and remaining stock.

"*We write a monthly demand letter, but in case of emergency, we will ask in the middle also.*" (AHW, Health Post, Makwanpur)

"*If the stock only remains for one month, then we demand medicine. Medicines are supplied every month.*" (Pharmacy Assistant, Hospital, Makwanpur)

**Theme 2: Unrealistic demand estimation.**    Participants expressed that they faced challenges estimating essential medicines needs, relying on previous months' consumption data and patient flow. The province government estimates the number of essential medicines to procure through the population, morbidity data and program basis.

"*We estimate the required amount of essential medicine based on population, morbidity, and program, but we do not have exact data of population. It is quite challenging to estimate morbidity data.*" (Logistic Officer, HLMC, Bagmati Province)

"*Mostly, we guess by seeing the previous year's Authorized Stock Level (ASL) and Emergency Order Point (EOP) data. Sometimes, patients demand more medicine than the previous year, but we cannot fulfil demand because of the limited stock of medicine. As a result, ASL and EOP get less.*" (Health Assistant, Health Post, Chitwan)

**Theme 3: Transportation.** Transportation challenges were reported, with participants from hilly areas emphasizing the difficulties of moving essential medicines during adverse weather conditions.

"*Municipality delivers essential medicines in Health Post by their office vehicle, and sometimes health post staff bring them while returning from the municipality. They (municipality officials) say that they do not have enough money for transportation services.*" (Health post-In-charge, Chitwan)

"*Sindhuli (district) lies in the hilly region. Most of the roads are difficult and damaged. So, we use tractors every season to transport medicines. In the winter season, Bolero jeep is used.*" (Health Post In charge, Sindhuli)

"*Sindhupalchowk is a disaster-prone area. Every year we face the problem of floods and landslides in the rainy season. For transportation of medicines, we carry half of the way in the bus, then we need to carry medicines to health post level. Porters are hired to carry medicines on their backs. Sometimes, Bolero Jeep and bikes are used for transportation. It is much more challenging to transport medicines in the rainy season.* (Health Assistant, Health Post, Sindhupalchowk)

## Domain 4: Shortage of Essential Medicines

**Theme 1: Shortage of essential medicines.** Participants reported experiencing varying level of shortages, with health posts near urban areas generally avoiding scarcities compared to those in rural areas. Most participants from health posts near city areas or municipality offices usually did not face a scarcity of essential medicines. However, health workers whose health posts were far from the municipality office or rural areas faced shortages of essential medicines. Family planning devices such as condoms, injectables and oral contraceptive pills, zinc tablets, amoxicillin, ciprofloxacin, and albendazole were in shortage in most health posts last year. Also, some participants mentioned the scarcity of medicines such as salbutamol, ranitidine, chlorpheniramine, vitamin B complex, Amlodipine and metformin in their health facilities. However, hospitals with pharmacies and where health insurance was operating did not face issues related to the scarcity of essential medicines.

"*Our health post is near the city area and has access to transportation facilities; Lalitpur metropolitan city itself is fully capable of supplying the required medicines and initiating procurement as per need. We never faced a shortage of medicines in our health facilities in the last one and half year.*" (Public Health Inspector, Health Post, Lalitpur)

"*We face shortage of essential medicines during rainy season because of lack of transportation and rise of water level. We also have shortage during the start and end of the fiscal year*" (Health Assistant, Health Post, Sindhupalchowk)

**Theme 2: Delay in receiving supply.** Service providers at the local level reported shortages and delays in supply from the municipalities, along with stock-outs at the municipality and health office levels.

"*Municipality offices do not supply essential medicines according to the demand. During the rainy season, we demand essential medicines for five months by filling out the form of LMIS, but they give little medicine which is not sufficient. Municipality offices do not procure*

*medicines on time, which is the main reason for the shortage. Also, sufficient budget is not allocated for essential medicine procurement.”* (Health Assistant, Health Post, Chitwan)

**Theme 3: Measures to prevent shortage.** Participants described various strategies to address shortages, including coordination with the municipalities borrowing medicines from neighboring health facilities and keeping a one-month stock.

"If we don't have medicines, we ask municipality office and nearby health facilities to provide those medicines” (PHC In-charge, Sindhuli)

*“We keep backup of essential medicines for one month before demanding.”* (Health Assistant, Health Post, Chitwan)

**Theme 4: Health workers' frustration.** Health workers expressed frustration due to patient complaints and provided concerns regarding the reduced patient flow during shortages.

*“Sometimes, patients scold us. Patients and their relatives get angry when expected medicines are unavailable from the health post level. They get angry and question us, asking why you don't go home and rest, why you are here in health post if you cannot provide free medicines.”* (Health Post In Charge, Chitwan)

*“Yes, we have to face patients first. We have experienced such problems (shortage of medicines) very often. Sometimes people say that staff are hiding and selling the freely available medicines and materials outside of health post.”* (Public Health Inspector, Health Post, Lalitpur)

Some participants said that the flow of patients in the health post level was decreasing compared to previous years due to the shortage of essential medicines.

*“Before, we checked up to 50–60 patients, but now, we check only 10–20 people. This is because there is no continuity of medicine supply. They (Patients) will come in health post but do not get the medicine than they think, why to go to the health post.”* (Health Assistant, Health Post, Makwanpur)

**Theme 5: Economic impact on patient.** The shortage of essential medicines led to economic impacts on low-income patients, who were unable to access free medicines, and resorted to private healthcare. Most health workers opined that low-income patients usually visit health posts because they do not have enough money to go to a private hospital or clinic. People were compelled to go to a private clinic or hospital when essential medicines were unavailable in health posts. Participants agreed that it would have an economic impact on those patients.

*“Poor people come for the health service in our health post. People prefer private hospitals or clinics when free essential medicines are unavailable from the health post level. When people don't have money, they can't afford the cost of private health facilities.”* (Health Assistant, Health Post, Makwanpur)

## Discussion

This study explored the underlying causes of essential medicine shortages in the public sector health facilities within Bagmati Province, Nepal. These identified barriers encompass a range

of challenges, with delays in procurement emerging as the major obstacle leading to stock-outs of essential drugs. Other barriers include insufficient storage space, inadequate training on LMIS, lack of IT resources and power backup, unrealistic demand estimation, limited transportation budget, and reliance on a manual inventory management system.

The study identified delays in the procurement process as a critical factor contributing to essential medicines shortages at service delivery points. To address this issue, we recommend developing a comprehensive procurement guideline by the provincial government, accompanied by relevant training for the personnel involved. Moreover, we advocate for timely demand submissions from service delivery points to the procurement units, fostering collaboration among local governments, province hospitals, and HLMC. The typology of medicine shortages presented in this study was consistent with findings from previous studies conducted in Pakistan [24], China [25], and Europe [26]. This alignment emphasizes the universal nature of challenges in medicine procurement and distribution, underscoring the importance of a coordinated and comprehensive approach. In LMICs, concerns about poor-quality medicines have been widespread [27–29], often prompting calls to strengthen the national medicine regulators. Surprisingly, this study found that quality monitoring of procured essential medicines is rigorously enforced at the province level, ensuring a minimum shelf life of 18 months. However, challenges arise at the local level, where the absence of clear procurement guidelines leads to the risk of acquiring near-expiry medicines and duplication of orders. The governance structure in place can address these issues through effective communication and coordination.

Furthermore, the study highlights the budgetary constraints, with participants reporting inadequacies in the budget for essential medicines. We recommend allocating a dedicated budget for essential medicine procurement at PHCs and health posts to mitigate this, ensuring consistent availability and accessibility. This strategy promotes positive health outcomes and alleviates the financial burden on patients seeking essential medicines from private pharmacies.

The inventory management challenges identified, particularly reliance on conventional registry methods, lack of resources, and inadequate training in e-LMIS, underscore the need for a skilled workforce and technology-friendly environments. Another barrier to inventory management was inadequate storage space and skilled human resources. Training health workers in logistics management or assigning individuals with basic logistics training can enhance efficiency and optimize healthcare service delivery. Prioritizing technology-friendly environments through regular e-LMIS training and sufficient IT resources is crucial, acknowledging potential resistance [24, 30].

Additionally, our findings align with global challenges in essential medicine procurement, as evidenced by studies from Ethiopia [31], Kenya [32], South Africa [33], and Uganda [34]. Abiye et al. (2013) identified barriers such as non-availability, high prices, inaccessibility, and ineffectiveness of drugs [31]. Fredrick and Muturi (2016) emphasized the lack of information and communication technology manipulation, uncoordinated supply chains, and inadequate staff qualifications affecting essential medicine availability [32]. Studies in Kenya [35], Malawi [36], South Africa [33] and Uganda [34] reported challenges related to delays in procurement processes, pharmaceutical supply chain management, staff inadequacy, and inadequate funding. These findings complement the barriers identified in our study, such as delays in procurement, inadequate storage space, and insufficient budgets.

Prioritizing a technology-friendly environment in health facilities through regular e-LMIS training and providing sufficient IT resources is recommended. The study conducted at Malaysian government hospitals pointed out that implementing electronic inventory management systems may face resistance due to negative mindsets and perceptions, making it difficult to introduce changes [37]. Therefore, it is essential to implement effective change management

strategies in the future to ensure a smooth transition to new technologies [37]. The limited storage space hindered the effective utilization of the e-LMIS system and the practice of FEFO in our study setting, highlighting the need for adequate storage space in health facilities. Assessment of the existing space in each health facility and allocation of space by the local governments would help to solve the problem of inadequate storage space. Temporary storage areas should be designated and maintained appropriately to ensure the physical integrity of drugs [38]. Addressing storage space inadequacies and implementing health emergency plans considering challenges such as COVID-19 can further fortify essential medicine management [39, 40]. Ensuring effective management of essential drugs and commodities is crucial for meeting the needs of service users [41]. However, this study found that untimely reporting from local governments complicates accurate monitoring of essential medicines supplies. The reliance on manual records at some health facilities and lack of confidence in electronic reporting contribute to inconsistencies in the essential drugs and commodities supply. To address these challenges, the provincial government should prioritize robust tracking and managing systems empowering local governments to make informed decisions about procurement and distribution.

## Strengths and limitations

In the broader context, our study contributes to understanding essential drug management during the transition of Nepal from a unitary to a federal health system. With a substantial sample size and diverse participant representation, the findings provide valuable insights for policymakers and health authorities seeking to enhance access to essential medicines at the sub-national level. Generalizing the findings from this study at the national level requires some caution. While the study has strengths, such as its comprehensive analysis, it also faces limitations due to the COVID-19 pandemic, which impacted physical interaction and participant engagement. Obtaining participants' time was hindered by mobile network disruptions, internet connectivity issues, interference with work schedules, and hesitancy to provide opinions. These limitations should be considered when interpreting the findings of the study.

## Conclusion

In conclusion, our study conducted in Bagmati Province, Nepal, underscores delay in procurement as a critical factor leading to essential medicine shortages. Addressing the identified barriers, including delays in procurement, quality monitoring, logistics management capacity, storage space, and IT resources, is imperative for effective essential medicine management in Bagmati Province. Addressing this issue necessitates the implementation of a comprehensive procurement guideline, training initiatives, and collaborative efforts among stakeholders. Budgetary constraints and challenges in inventory management further emphasize the importance of dedicated budgets, skilled personnel, and technology-friendly environments. By implementing the recommended measures and learning from global experiences, the province can navigate its transition to federalism, ensuring improved procurement policies and addressing management issues to enhance access to essential drugs. Our research aligns with global challenges in essential medicine procurement, highlighting the universal nature of these issues. While offering specific insights for Nepal, our study recommends practical strategies to enhance pharmaceutical supply chain management. To successfully navigate the transition to federalism, the province must adopt improved procurement policies and address management issues, ultimately enhancing access to essential drugs. Our findings call for coordinated efforts, training programs, and strategic investments to improve the overall availability and accessibility of essential medicines. This comprehensive approach is necessary for ensuring the

effectiveness of essential medicine management in Bagmati Province and can serve as a model for addressing similar challenges globally.

## Supporting information

**S1 Text. Interview guide.**
(DOCX)

## Acknowledgments

The authors thank Mr. Mahesh Nath, Ms. Rajina Adhikari, Mr. Shishir Khanal, Mr. Sudip Bahadur Pulami Magar, Ms. Ranju KC, Mr. Aman Raj Pariyar, and Ms. Sarala Nesur for their support in data collection and Mr. Kiran Paudel for supporting the data management process. We are also thankful to study participants from Bagmati province for their time despite their engagement in responding to COVID-19 outbreak.

## Author Contributions

**Conceptualization:** Basant Adhikari, Kamal Ranabhat, Pratik Khanal, Manju Poudel, Sujan Babu Marahatta, Saval Khanal, Sunil Shrestha.

**Data curation:** Kamal Ranabhat.

**Formal analysis:** Basant Adhikari, Kamal Ranabhat, Pratik Khanal, Sunil Shrestha.

**Funding acquisition:** Basant Adhikari, Kamal Ranabhat, Pratik Khanal, Sunil Shrestha.

**Investigation:** Basant Adhikari, Kamal Ranabhat, Pratik Khanal, Sunil Shrestha.

**Methodology:** Pratik Khanal, Saval Khanal, Vibhu Paudyal, Sunil Shrestha.

**Project administration:** Basant Adhikari, Kamal Ranabhat, Pratik Khanal, Manju Poudel, Vibhu Paudyal, Sunil Shrestha.

**Resources:** Basant Adhikari, Manju Poudel, Sujan Babu Marahatta.

**Software:** Basant Adhikari, Kamal Ranabhat, Pratik Khanal, Sujan Babu Marahatta.

**Supervision:** Basant Adhikari, Kamal Ranabhat, Pratik Khanal, Sujan Babu Marahatta, Saval Khanal, Vibhu Paudyal.

**Validation:** Basant Adhikari, Sunil Shrestha.

**Visualization:** Basant Adhikari, Kamal Ranabhat, Sunil Shrestha.

**Writing – original draft:** Kamal Ranabhat, Pratik Khanal, Sunil Shrestha.

**Writing – review & editing:** Basant Adhikari, Kamal Ranabhat, Pratik Khanal, Manju Poudel, Sujan Babu Marahatta, Saval Khanal, Vibhu Paudyal, Sunil Shrestha.

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
