## [Decision Letter · Decision Letter 0]

5 Jan 2024

PGPH-D-23-02026

Shortage of Essential Medicines and Challenges with the Procurement Process in Public Health Facilities in a low resource setting: A Qualitative Analysis from Nepal

Dear Dr. Shrestha,

Thank you for submitting your manuscript to PLOS Global Public Health. After careful consideration, we feel that it has merit but does not fully meet PLOS Global Public Health’s publication criteria as it currently stands. Therefore, we invite you to submit a revised version of the manuscript that addresses the points raised during the review process.

We look forward to receiving your revised manuscript.

Kind regards,

Vanessa Carels

Staff Editor

Journal Requirements:

2. Please provide separate figure files in .tif or .eps format only and remove any figures embedded in your manuscript file. Please also ensure all files are under our size limit of 10MB.

3. In the online submission form, you indicated that "The datasets used and/or analysed during the current study are available from the corresponding author upon reasonable request". All PLOS journals now require all data underlying the findings described in their manuscript to be freely available to other researchers, either 1. In a public repository, 2. Within the manuscript itself, or 3. Uploaded as supplementary information.

Additional Editor Comments (if provided):

Reviewers' comments:

Reviewer's Responses to Questions

**Comments to the Author**

1. Does this manuscript meet PLOS Global Public Health’s publication criteria? Is the manuscript technically sound, and do the data support the conclusions? The manuscript must describe methodologically and ethically rigorous research with conclusions that are appropriately drawn based on the data presented.

Reviewer #1: Yes

Reviewer #2: Partly

Reviewer #3: Yes

2. Has the statistical analysis been performed appropriately and rigorously?

Reviewer #1: N/A

Reviewer #2: No

Reviewer #3: Yes

3. Have the authors made all data underlying the findings in their manuscript fully available (please refer to the Data Availability Statement at the start of the manuscript PDF file)?

Reviewer #1: Yes

Reviewer #2: No

Reviewer #3: Yes

4. Is the manuscript presented in an intelligible fashion and written in standard English?

Reviewer #1: Yes

Reviewer #2: No

Reviewer #3: Yes

5. Review Comments to the Author

Reviewer #1: Article acceptable once all quantitative language is corrected in particular use of questionnaire should be replaced with interview guide.

Reviewer #2: The title mentioned " Shortage of essential medicines", kindly provide necessary data justifying your view point as it currently lacking in your manuscript.

The enrollment of health facilities is not very convoluted. kindly make it simpler also include data such as number of each types of health facilities (PHC, CHC, Hospital etc.) available in each district, number of each types of health facilities selected from each district and method of selection of health facilities.

Out of total 59 interviews conducted, please elaborate the designation of interviewees. present data in terms of percentage

How informed consent was taken ( via phone , google form etc.) ?

Results are not written clearly. number of questions under each theme and response for each question by different interviewees should be discussed in results.

Reviewer #3: The article is good but requires some improvement and clarifications based on review I have attached in here including the background, and the discussion sections which requires more revision. This is a qualitative research and therefore did not require statistical analysis.

6. PLOS authors have the option to publish the peer review history of their article (what does this mean?). If published, this will include your full peer review and any attached files.

**Do you want your identity to be public for this peer review?** For information about this choice, including consent withdrawal, please see our Privacy Policy.

Reviewer #1: No

Reviewer #2: No

Reviewer #3: **Yes: **Dr Kezia Njoroge

---

## [Decision Letter · Decision Letter 1]

27 Mar 2024

Procurement process and shortages of essential medicines in public health facilities: A qualitative study from Nepal

PGPH-D-23-02026R1

Dear Dr. Shrestha,

We are pleased to inform you that your manuscript 'Procurement process and shortages of essential medicines in public health facilities: A qualitative study from Nepal' has been provisionally accepted for publication in PLOS Global Public Health.

Best regards,

Julia Robinson

Executive Editor

Reviewer Comments (if any, and for reference):

Reviewer's Responses to Questions

**Comments to the Author**

1. If the authors have adequately addressed your comments raised in a previous round of review and you feel that this manuscript is now acceptable for publication, you may indicate that here to bypass the “Comments to the Author” section, enter your conflict of interest statement in the “Confidential to Editor” section, and submit your "Accept" recommendation.

Reviewer #1: All comments have been addressed

Reviewer #3: All comments have been addressed

2. Does this manuscript meet PLOS Global Public Health’s publication criteria? Is the manuscript technically sound, and do the data support the conclusions? The manuscript must describe methodologically and ethically rigorous research with conclusions that are appropriately drawn based on the data presented.

Reviewer #1: Yes

Reviewer #3: Yes

3. Has the statistical analysis been performed appropriately and rigorously?

Reviewer #1: N/A

Reviewer #3: N/A

4. Have the authors made all data underlying the findings in their manuscript fully available (please refer to the Data Availability Statement at the start of the manuscript PDF file)?

Reviewer #1: Yes

Reviewer #3: Yes

5. Is the manuscript presented in an intelligible fashion and written in standard English?

Reviewer #1: Yes

Reviewer #3: Yes

6. Review Comments to the Author

Reviewer #1: suggest author remove a qualitative study in the title

.....Nepal public health facilities sound better

Reviewer #3: (No Response)

7. PLOS authors have the option to publish the peer review history of their article (what does this mean?). If published, this will include your full peer review and any attached files.

**Do you want your identity to be public for this peer review?** For information about this choice, including consent withdrawal, please see our Privacy Policy.

Reviewer #1: No

Reviewer #3: **Yes: **Kezia Njoroge
